# Pedodiversity of Subboreal Ecosystems under Contrasting Geogenic Factors (Case Study of Samarskaya Luka, Middle Volga Region, Russia)

**Evgeny Abakumov**

Department of Applied Ecology, Saint-Petersburg State University, 16 line 26 Vasilyevskiy Island, Saint Petersburg 199178, Russia; e_abakumov@mail.ru or e.abakumov@spbu.ru

**Abstract:** The soils of uplands and partially isolated landforms located in the central Volga region were investigated in terms morphology, diversity and geogenic factor specificity. The Samarskaya Luka area is characterized by extreme spatial inhomogeneity and contrast of geogenic (lithological and topographic) conditions, and at the same time is located on the ecotone-transitional border of several natural zones. This territory is a part of two federal protected areas because of its peculiar nature. It is established that the diversity of geogenic (geological and topographic) conditions leads to the differentiation of the soil cover. Key soil types are presented by Leptosols, Cambisols, Retisols, Chernozems, Fluvisols and Technosols. By the example of the recognized upland macrolandscape, it is shown that the usual gradual course of soil zonation significantly changes due to the transformation of the soil-forming potential of the environment. The obtained data can be used for further regional verification of soil taxonomy. In addition, since the role of parent materials increases as they move from the west to the east of the East European Plain, parameterization of the role of this process becomes particularly important. Information about the diversity of chemical parameters and the granulometric composition of the key soil types of the studied area is also given.

**Keywords:** soil; pedodiversity; parent materials; geogenic conditions; subboreal environments; middle Volga region



## 1. Introduction

Russian soil science has a rich history, starting from the publication by V.V. Dokuchaev of his book "Russian Chernozem" [1]. After which about 1/6th of the land surface has been covered over the past 100 years by soil and geographic research of the Dokuchaev school. It was here, on the Russian or East European Plain, that V.V. Dokuchaev discovered the phenomenon of latitudinal soil zonality [1]. Although zonality is traced in the European and Siberian parts of the Russian Federation, the character of zonal series in different continental conditions is very different [2–4]. Geogenic factors include both geological and topographical ones, these factors are closely related to each other, thus the geology determines, in many ways, the forms of relief, and the surface topography affects the redistribution of weathering products on the slope. The role of soil-forming parent materials in the diversification of soil formation and soil evolution was highlighted by the classics of world soil science [5–7]. If we analyze the influence of geogenic factors on zonality, then even within the East European Plain, the character of zonal distribution of soils will be different for the western and eastern parts [5]. If we add to this the substantial activity of rivers in transforming the regional climate and lithology, the course of the latitudinal zonality will be substantially reversed and will differ from the ideal Dokuchaev model. Thus, E. Kolomyts proved [8] that the boreal ecotone and the corresponding zonal and subzonal series of soils, vegetation zones and landscapes are largely related to the activity of the Volga River, which modifies climatic and lithological conditions. Within the Volga

macro ecotone, the border between natural zones is broad and not sharp, soils of one zone appear in another due to the action of climatic and lithological factors.

The specificity of soils of the eastern parts of the East European Plain has been noted by many researchers. The peculiarity of the soils of the Volga uplands was pointed out by I.S. Urusevskaya [9], who showed that they are characterized by a lower profile thickness due to the specificity of parent materials (deluvium, loess deluvium) compared to similar soils of the Srednerusskaya Upland, formed on loess-like loams. The same can be said about syrt chernozems in the Saratov and Orenburg regions (syrt are defined as an upland of complicated marine-loess genesis, typical for the south of the European Russian part and Kazakhstan). East of the Volga River the specificity of geogenic factors is manifested more and more, modifying the usual zonal course of soil distribution. This is also superimposed on the continental climate, as well as the appearance of red-colored parent material-derivatives of Permian deposits and sediments of ancient transgressions of the Caspian Sea [10]. Thus, the soil zonality and soil diversity of the east and west territories of the East European Plain divided by the Volga River are quite different. Additionally, one of the reasons for this is the feature of the spatial distribution of soil-forming rocks on the territory of the East European Plain [11]. Samarskaya Luka is located on the border of the western and eastern parts of central European Russia. Previously, many field researches have been conducted on soil morphology, routine chemistry and ecological functions of soils of Samarskaya Luka [4,10,12]. That is why it may be a good model object for clarification of the role of geogenic factors in spatial soil distribution and formation of increased pedodiversity. Thus, the aim of our study was to evaluate dependence of soil diversity on geogenic factors in the central part of the subboreal climatic belt. To achieve this aim the following objectives were formulated:

(1) To describe the diversity of soil parent materials and to provide their chemical and mineralogical composition;
(2) To analyze the role of geogenic factors in geographical soil pattern on Samarskaya Luka;
(3) To evaluate the dependence of soil diversity from lithological inhomogeneity of territories.

## 2. Materials and Methods

### 2.1. Regional Settings

Samarskaya Luka is located in the eastern part of the Privolzhskaya Upland, which, along with the Srednerusskaya Upland, is the most important habitat of forest-steppe and steppe landscapes in automorphic positions. Samarskaya Luka is washed by the waters of the Volga River, joined to the Privolzhskaya Upland by a small isthmus [10]. This elevation is occupied by two specially protected territories of federal level (Figure 1). One of them, the most strictly protected, is the Zhigulevsky State Reserve. The second territory with a less strict protection regime is the "Samarskaya Luka" National Park.

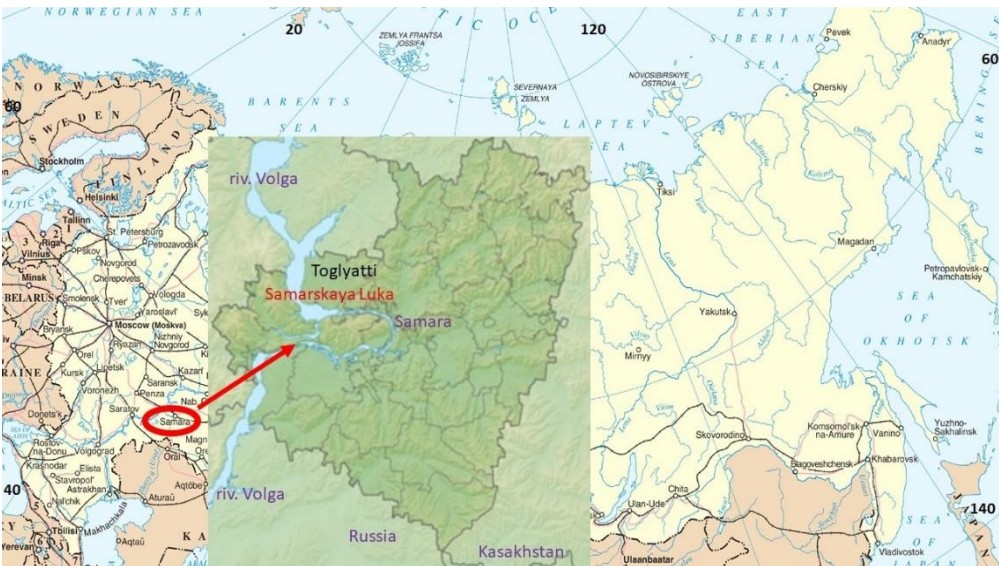

**Figure 1.** Insert map of Samarskaya Luka, Povolzhye region and Russia (https://www.infoplease.com/atlas/asia/russia-map, accessed on 10 October 2022).

### 2.2. Climate and Biota

The climate of the Samarskaya Luka peninsula is sharply continental [13]. Sharp climatic differences are amplified by the influence of the relief. The average annual amount of precipitation is 566 mm for the weather station at Bakhilova Polyana and 610 mm for the Sosnovy Solonets weather station. The average annual air temperature is +4.8 °C and +4.5 °C; average air temperature in January is −10 °C and −12 °C, in July it is +20 °C and +21 °C for these stations, respectively. The presence of the water mirror of the Kuibyshev reservoir contributes to some softening of the climate on the slopes of the Zhigulevsky mountains with northern exposure. The territory of Samarskaya Luka is part of the East European forest-steppe province and Eurasian steppe province [7]. The northern part belongs to the broad-leaved, mixed forests subzone and the southern part to the forest-steppe zone. zone of mixed forests, while the southern part belongs to the forest-steppe zone. The boundary between them is situated to the south of the Shiryaev valley. The territory of the Samarskaya Luka consists of 55% forest. In the reserve, the forests cover 92% of its area. A specific peculiarity of Samarskaya Luka is the presence of Europe's rarest relict pine woods on limestone with steppe vegetation under the forest canopy and stony steppes with shikhans (eroded elevations of the limestone slopes or limestone ridges).

### 2.3. Geogenic Conditions and Landforms

Samarskaya Luka is a peninsula, which is washed by the waters of the Volga River to the north, east and south [13–15] Volga, and from the south by the waters of the Usa River; only a narrow strip of land (Rus. "Perevoloki"—ship translocation land) connects it with the eastern part of the Volga uplands [14]. The Samarskaya Luka Upland is one of the most contrasting parts of the European boreal ecotone of the Russian Plain [4,15,16]. The maximum extent of this territory from west to east is almost 110 km, and 33 km from north to south. It rises as a high forested island among the black-soil-steppe landscapes of this part of the Volga uplands. The highest points are in the northern part of the Samarskaya Luka massif—the Zhigulevskiye Mountains—where the reserve is located. Within Samarskaya Luka there are several. The Zhiguli or Zhiguly ridges are a narrow (2–3 km wide) forested and rugged strip of land located along the northern edge of the bend; the absolute heights dominate. A wide, 2–3-km-wide dissected strip of woodland located along the northern edge of the bend; the absolute heights prevail; the forest plateau with the highest elevation of 381 m above sea level is a panorama of the Samara Region. Greater than 300 m—forested plateau, occupying the eastern half of the Luka, absolute heights of 200–250 m; forest-steppe

dissected plateau—the western part of the Luka, characterized by absolute heights of up to 200 m, denudation valleys of the post-Dneprovian age (Bakhilova, Shiryaevskaya, etc.). Soil parent materials in Samarskaya Luka are represented by Pre-Quaternary and Quaternary sediments [10]. Pre-Quaternary parent materials: Carboniferous and Permian limestones; Carboniferous and Permian dolomites; Jurassic clays, heavy and medium loams, sandy loams and sands, Permian gypsums, marine clays of the Kinelsky and Akchagylsky stages of the Neogene, Paleogene secondary quartzite and sandstones. The Quaternary parent materials list is given in Table 1.

**Table 1.** Quaternary sediment and related relief forms.

| Type of Relief | Type of Quaternary Parent Materials |
| --- | --- |
| Zhiguli Mountains the tops and upper parts of the slopes | Eluvium of limestones and dolomites of the Carboniferous and Permian |
| Zhiguli Mountains the tops and upper parts of the slopes of the Eastern part of Samarskaya Luka | Eluvium of gypsum |
| Middle and lower Slopes of Zhiguli ridges, intermountain valleys | Colluvium in the mountainous part |
| Central denudation plateau of Samarskaya Luka | Eluvium of Jurassic clays, loams, sands; |
| Southern denudation plateau of Samarskaya Luka | Eluvium of the Akchagyl transgressive sand and clay series; |
| Eastern-southern denudation plateau of Samarskaya Luka | Eluvium of Neogene greenish transgressive clays |
| South-western part of plateau of Samarskaya Luka | Loess-like loams |
| Denudation valleys of late Pleistocene | Loess type slope deluvium |
| Mountainous, plateau, slope | Deluvium of various sediments and of mixed composition |

Thus, the diversity of parent materials is quite high and landscapes of the Samarskaya Luka are quite different in terms of chemical composition, mineralogy and texture. This could be considered as an essential predictor of the pedodiversity formation.

### 2.4. Field Survey, Soil Diagnostics

Field studies were conducted from 2001 to 2019 in the course of expeditions by St. Petersburg State University and the Institute of Ecology of the Volga basin in the territory of Zhigulevsky State Reserve, named after I.I. Sprygin and Samarskaya Luka National Park and adjacent territories. Soil-topographic profiles through all key landforms of the Samarskaya Luka Upland were formed. In total, more than 300 soil sections were described. This paper summarizes the results of many years of soil research in Samarskaya Luka. Laboratory studies included traditional research methods: determination of the basic physical and chemical characteristics of soils, their granulometric and mineralogical composition.

### 2.5. Laboratory Methods

Soil samples were ground and passed through a 2-mm sieve to obtain fine earth. All the analyses were conducted on the fine earth after weighing the skeletal fraction. The C and N total content in plant materials were determined using an element analyzer (Euro EA3028-HT Analyser, Pavia, Italy). The pH levels were correspondingly measured potentiometrically with the ratio of soil and water of 1:2.5 and 1:25 for mineral and organic horizons. The carbonate content was determined by wet method with addition of 1 M hydrochloric acid and titrimetric residue of acid of 0.5 M solution of NaOH. Soil texture were determined on the basis of the classical sedimentation method. Clay mineralogy

of parent materials was determined. From the samples of fine-grained gravel, distilled water was used to extract the clay fraction with the size of mechanical elements less than 0.002 mm. From this fraction sample preparations were made by sedimentation on a substrate of quartz glass. The samples were observed with a DRON-6 X-ray diffractometer with Co$\alpha$-monochromatic radiation, wavelength $\lambda = 1.79021°$A, voltage U = 35 kV, current I = 25 mA. I = 25 mA and in step-by-step mode with 0.02° step. For revealing the labile component, preparations of the clay fraction of samples were saturated with ethylene glycol. For crystallochemical characteristics of phases, preparations of samples were heated in a muffle oven at temperatures of 350 °C and 650 °C for one hour. The obtained spectra were processed using the PDWin-4 software package. The phases were identified using the JCPDS index card. For semi-quantitative determination of minerals, we used the method of Yu. S. Dyakonov [17].

## 3. Results and Discussion

### 3.1. Characterization of Parent Materials

The most common Quaternary sediments are represented by several genetic groups: eluvial formations of watersheds and eluvial-deluvial formations of mountain areas (skeletal eluvium, red-brown eluvial clays, porous clays of gullies, rock-fall formations); deluvium (dark or reddish-brown loams with columnar loams), pillar-shaped loams, thickness of deluvium varies from 1 to 15 m; aeolian formations (sands); ancient valley sediments of various genesis and age (loess-like loams, loess-like deluvium, alluvium of temporary watercourses); and alluvium of Volga and Usa valleys. Brief analytical characteristics of the most widespread parent materials are given in Table 2.

**Table 2.** Key characteristics of soil parent materials.

| Type of Parent Material | Type of Dominated Soil | Content of Carbonates, % | pH in Water of Parent Material | Content of Clay, % | Mineralogical Content of Clay Fraction |
|---|---|---|---|---|---|
| Eluvium of limestones | Leptosols Calcaric | 30–80 | 7.6 | 25–30 | Smectite-kaolinite-illite |
| Deluvium of bedrock mountain slopes | Cambisols | 12–15 | 7.3 | 65–70 | Smectite-illite-kaolinite |
| Deluvium bedrock Plateau | Retisols, Umbric Retisols | 5–9 | 7.0–7.5 | 50 | |
| Jurassic clay eluviums | Retisols Albic | 0 | 5.0 | 30–60 | Illite-kaolinite at very small smectite admixture |
| Loess-like deluvium of denudation valleys | Phaeozems Luvic | 10–12 | 7.0–8.3 | 40–50 | Smectite-illite-kaolinite |
| Proluviums of denudation valleys | Phaeozems Luvic | 0–4 | 6.0–7.2 | 35–50 | Smectite-kaolinite-illite |
| The Akchagyl clays | Phaeozems Petrocalcic Calcaric | 18–27 | 8.0–8.5 | 45–55 | Smectite-illite-kaolinite |

It is noteworthy that amorphous silica in the silty fraction occurs only in the rocks from which texturally differentiated soils are formed. Thus, derivates of Jurassic parent materials are favorable to podzolization and formation of Albeluvisols (Retisols)—soils with evident features of redistribution of clay fraction in vertical profile scale. In numerous parent materials of Samarskaya Luka one can link the presence of the mineral smectite with a labile crystal of clay minerals. This determines many of the rheological and water-physical properties of soils. In particular, this factor can cause vertisolization or so-called soil mass cohesion

(slitization). Akchagyl sediments and other ancient clays are often swollen and could be found in Povolzhye region as well as slitisized (vertic) soils [18]. The presence of vertic in soils from Samarskaya Luka was not fixed by Nosin [4], thus the question of whether future research is required to understand if smectite in inherited from parent materials or currently formed mineral in theses soils is justified. The majority of parent materials of Samarskaya Luka contains solid primary carbonates, thus the soils, originated from these materials, could be classified with addition of the verifier "Calcaric" according to WRB-2015 [19]. At the same time, while the Samarskaya Luka peninsula is located in a forest-steppe subboreal region [4], there is enough gravimetric water in the soil for seasonal formation of vertical preferential flow and further development of eluviation processes. That is why, much of the soil has Luvic properties expressed in soil morphology and chemistry. The clays and loams of the region are denser than loesses and loess-like sediments of Srenderusskaya Upland. This reason was mentioned by Urusevskya et al. [9] as a factor for different depths of Umbric Luvisols profiles if one compares Srenderusskaya and Privolzhskaya uplands. Calcareous parent materials with inherited lithogenic carbonates are parent materials for the formation of Leptosols on the tops and highest slopes of the Jiguli ridges in Samarskaya Luka. The pH values of the parent material are not higher than 8.5 units, thus there is no any evident reason for active soil salinization. Thus, the soil cover of this peninsula is quite different from the soil cover of adjacent territories—Nizmennoye Zavolgye, Vysokoye Zavolgye and Privolgzhskya Upland. One of the factors, regulation of pedodiversity on Samarskya Luka, is spatial inhomogeneity of geogenic factors—topography and relief. The floodplains of the Samarskaya Luka have survived on a very limited scale. This is due to the construction of the Zhigulevskoye and Saratovskoye reservoirs. Nevertheless, the Rozhdestvenskaya floodplain and the floodplain near the settlement of Bakhilova Polyana have been preserved. Here, the lithological matrix is represented by layered alluvial loams, alluvial and ancient aeolian sandy loams. The relief of floodplains is differentiated and developed. Up to seven types of alluvial soils are formed in floodplain catenas, as well as spodic soils on different-height alluvial terraces.

*3.2. Soil Diversity*

Soil distribution and diversity are strongly affected by topography. Some examples of landscapes and biotopes are given in Figure 2. The distribution of soils across the main types of landscapes can be represented as follows.

The northern mountainous area belongs to the broad-leaved forest zone. Soils distribution is affected by geogenic conditions: Calcaric Leptosol (top parts of ridges), Cambisols (middle parts of northern slopes of mountains) and Luvic Chernozems (in valleys and gorges). The accumulation of talus material is observed at the bottom of the slope, and coastal abrasion is observed near the river bed.

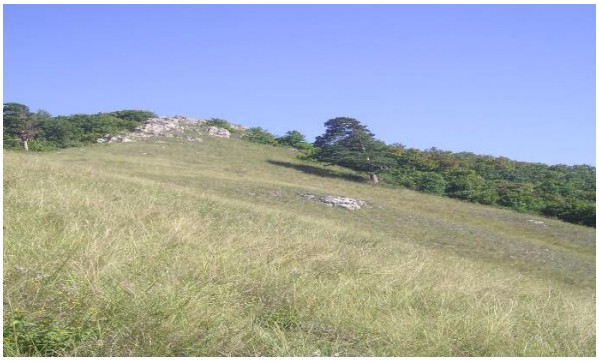

**1**

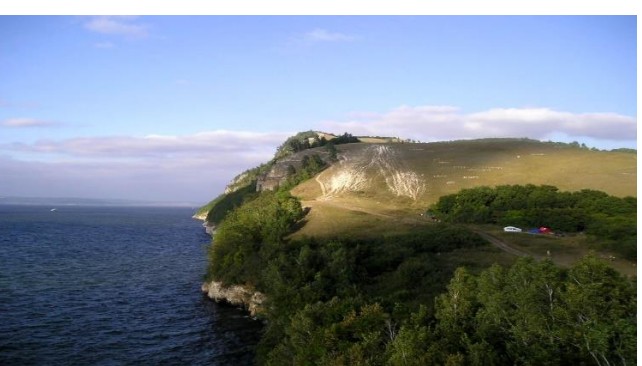

**2**

**Figure 2.** *Cont.*

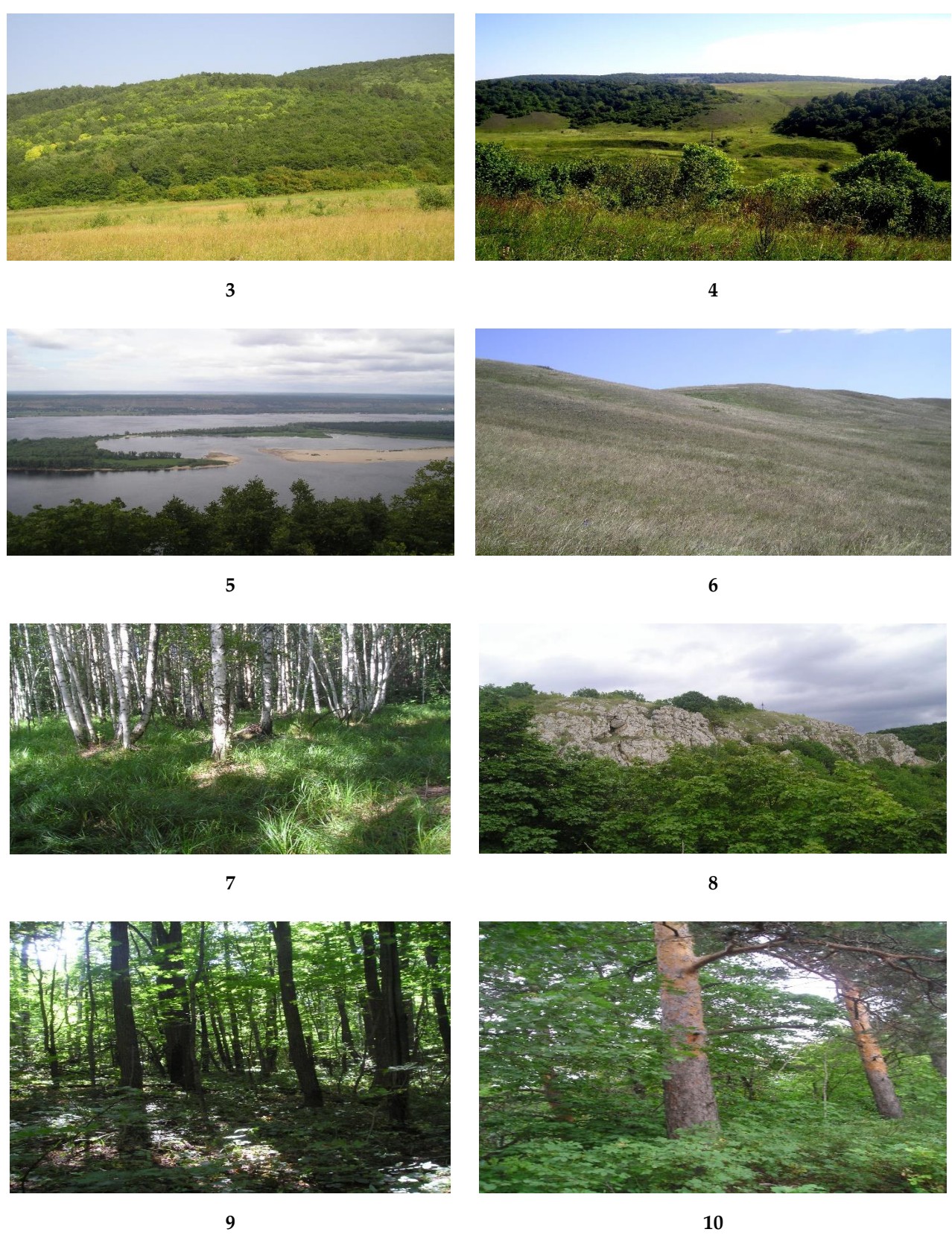

**Figure 2.** Key landforms and biotopes of Samarskaya Luka. **1**—steppe slopes of the Zhigulevsky mountains "Shikhany" (N 53-26-073, E 49-40-809, h = 240 m), **2**—slopes of the Zhigulevsky mountains, steppe landscapes disturbed by recreation (N 53-19-473, E 49-15-193, h = 80 m), **3,4**—early Holocene denudation valley slope, composed of deluvium, forest-steppe vegetation (N 53-19-384, E 049-50-508,

h = 167 m), **5**—alluvial sandy islands of the Volga River in the upper part of the Saratov reservoir, floodplain forests and meadows (N 58-67-655, E 32-78-277, h = 30 m), **6**—true steppes of the southern part of Samarskaya Luka, located on denudation slopes covered by eluvium and deluvium (N 53-28-129, E 49-60-991, h = 110 m), **7**—birch forests on Jurassic loam on the Samarskaya Luka plateau (N 53-27-444, E 49-95-434, h = 70 m), **8**—oak (*Quercus petraea*) forests on skeletal thin soils (N 53-26-068, E 049-41-786, h = 270 m), **9**—linden forest (*Tilia cordata*) on the Samarskaya Luka plateau on loess-like loams (N 53-19-026 E49-15-742, h = 90 m), **10**—pine forests (Pinus sylvestris) on dry mountain slopes on the Rendzic Leptosols (N 53-26-024, E 049-27-811, h = 220 m).

The northern upland area—transition of mountainous part to plateau—is covered with broad-leaved forests. Soils: Calcaris Leptosol and Luvic Chernozem are typical for places with dominance of skeletal eluviums and fine weathered deluviums of the corresponding uplands. Hilly plains, plateau-like uplands and valleys occupy most of the Zhigulevsky Reserve.

The central area of Samarskaya Luka plateau is a deluvial-denudation late Pleistocene relief with predominance of forests and presence of areas of steppe meadows. The area is distinguished by high diversity of soils: Sod Podzols, Histic Podzols, Stagnic, Umbric Retisols, Folic Retisols, Luvic Chernozems and Calcaric Chernozems. Podzols are located on Jurassic clays and their derivates. Luvisols are typical for deluviums of various clayey textured substrates. Chernozems are located on loess-like parent materials of loam texture with low percentage of carbonates. Thus, diversity of soils is first connected with presence of various soil-forming substrates (differences of soils at section and type level) and heterogeneity of relief (subtypes).

The southern region of flat slightly articulated plains, with the least changed soil-forming rocks, such as parent materials preserved after the Akchagyl sea transgression (3.4–1.8 years B.P.). Vegetation is represented by steppe meadows and areas of feather-grass steppes. Chernozems are widespread in the area, normally formed on loess-like loams and clays. The loess features (blocky structure, accumulation of fraction of 0.005–0.001 mm diameter) are pronounced only in 0–30 cm layers. Chernozems demonstrate features of secondary carbonates accumulation in the BCA horizon. Small separate polypedones are presented by chernozems, where the middle part of the profile is blue-green in color inherited from Akchagyl clays. The protection status of the south part of the peninsula is less strict in comparison with northern and middle parts belonging to the Jigulevskiy reserve while the southern part belongs to the National Park. Thus, agricultural practices are possible in this area and arable chernozems appear here.

The region of denudation plains (Shiryaevsky gorge, Otvazhninskaya, Morkvashinskaya and Bakhilova valleys, Gavrilova Polyana gorge) is represented by types of Luvic Chernozem and Umbric Retisols. Some Retisols demonstrate second humus horizons of illuvial origin. These valleys are relatively evenly distributed across the central part of Samarskaya Luka.

The floodplain area of the Volga and Usa rivers is represented predominantly by Fluvisols of various morphology and genesis. Due to the construction of reservoirs in the second half of the 20th century, floodplains and associated soils in the middle and lower reaches of the Volga almost did not survive. They were either flooded or subjected to substantial river bank abrasion. Therefore, the investigated soils are of special interest as examples of floodplain soil formation and in terms of special soil protection.

The region of anthropogenic soil formation and lithogenesis is distributed locally, in fragments and mainly represented by open-cast mines. The largest of them are located on limestone quarries. There are also quarries for Jurassic clays and sands. Several sulfur mines are revegetated by forest stands. The soil of the heaps is presented by primary Entisols with 10–20 cm solum depths.

### 3.3. Soil Morphology

Calcaric Leptosols vary greatly depending on location, parent materials and degree of their alteration. Soil sections are given on Figure 3. All profiles of Leptosols consist of humus AU Umbric horizons, transitional AC layers with high content of stones and calcium carbonate. Leptosols are sublayered by dense rock of the dolomitized limestone (Figure 3(1,2)) or limestones with admixture of gypsum (Figure 3(3)).

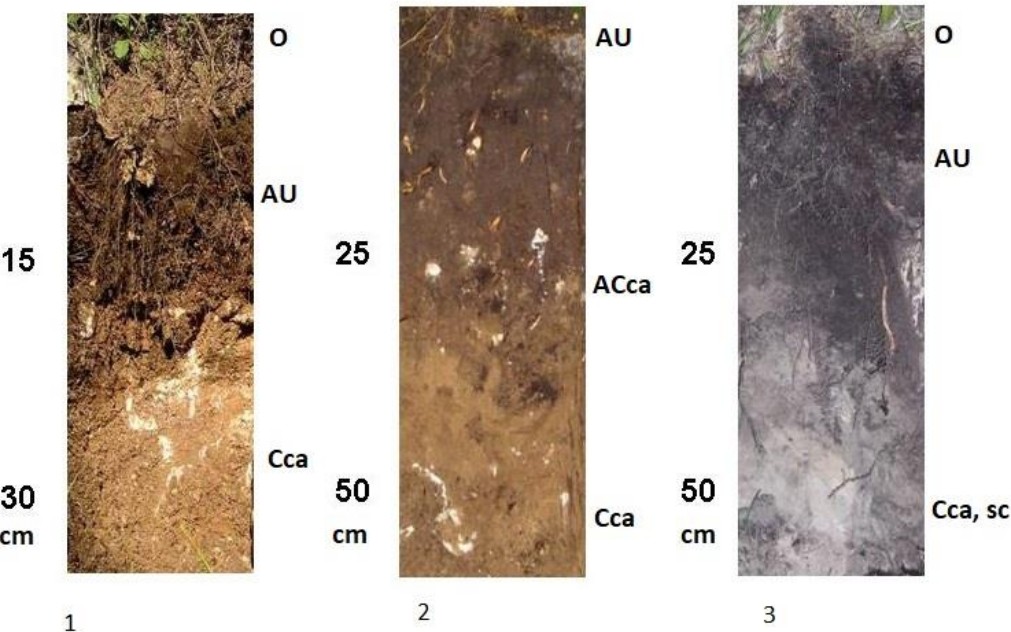

**Figure 3.** Calcaric Leptosols. **1**—Leptosol, hyperskeletic, on the top of limestone ridges, **2**—Leptosol, with weathered fine earth and transitional ACca horizon, on eluvium of limestone, **3**—Gypsic Leptosol.

The northern slopes of the Jiguli ridges are occupied by various Cambisols. The reason for their formation is increased humidity of air of ecosystems faced by the Volga River and Kuibyshevskoye water reserve. Cambisols have sharp interchange of texture in the middle part of the profile, which is reflected in Spanish verb "cambio"—to change [19]. This change is in situ, but not due to Luvic processes as with Retisols. That is why Cambisols are soils with very intensive in situ weathering. Hyperskeletic Cambisols (Figure 4(1)) occupy the highest parts of the slopes and merged with Leptosols, as seen in Figure 3. Middle parts of slopes with deeper mantle slope sediments are covered by well-developed, intensively leached Cambisols of brownish color (Figure 4(2,3)).

The diversity of Retisols (Figure 5) is also quite high. The reason for high variability of Retisol morphology is spatial variance of parent materials. Thus, the much leached and vertically differentiated profiles are formed on debris of Jurassic clays (Figure 4(1,2)). These soils could be classified as Podzolic soil according to the Russian soil taxonomy [20]. They demonstrate illuvial argillic horizons enriched by clay. Another type of Retisol is gray forest soils with Umbric horizons, dark-colored profiles with a less pronounced illuvial layer. These soils are located on more rich and alkaline parent materials of Holocene deluviums of loamy and clay texture. Thus, the lithogenic factor is responsible for pedodiversity in this case as well.

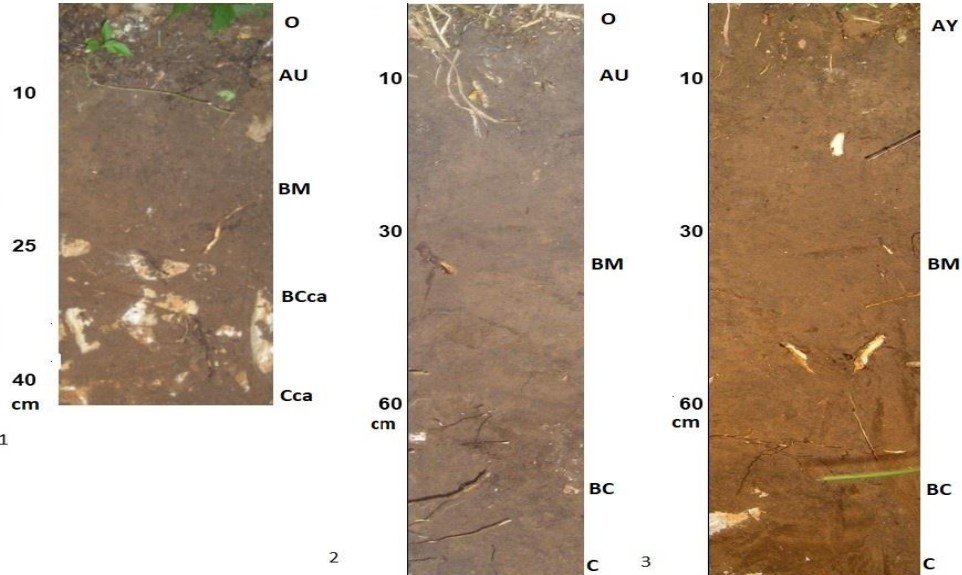

**Figure 4.** Cambisols of Jiguli ridges. **1**—Cambisol on skeletal eluvium of limestones, **2**—Cambisol on slope clay deluvium, **3**—Cambisol on valley clay textured sediments.

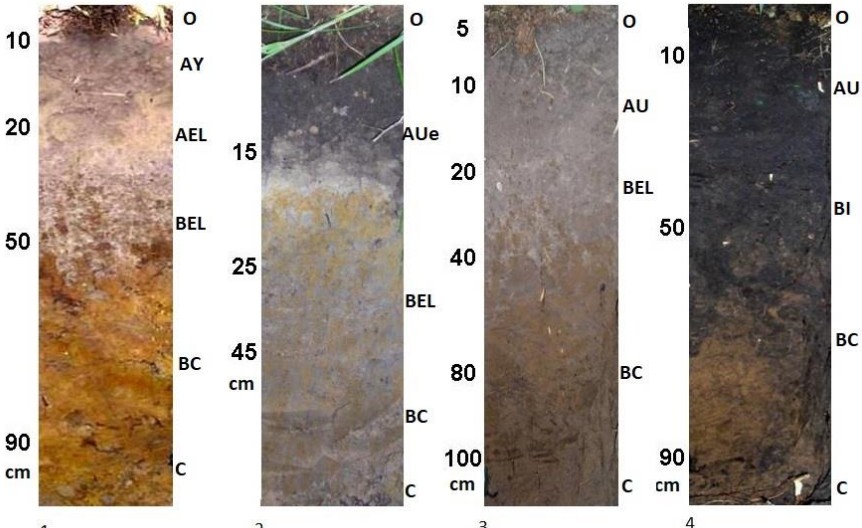

**Figure 5.** Retisols of Samarskaya Luka. **1**,**2**—Retisols with well-developed eluvial-illuvial features, northern part of plateau, **3**,**4**—Retisols with dark humus horizon and less developed eluvial differentiation, southern part of plateau.

The soil sections of Chernozems are provided in Figure 6. The most typical Luvic (Figure 6(1)) and Calcaric (Figure 6(2)) Chernozems are formed of slightly loessed loams with low and high percentages of carbonates, inherited from parent materials. Thus, the degree of Luvic processes depends on the carbonated content in the fine earth. The Chernozems of Samarskaya Luka have a less deep mollic layer of humus enriched soil in comparison with soils of Privolzhskaya and Srenderusskaya uplands, which is in good correspondence with data of Urusevskaya [6] who compared the development rate of Retisols in eastern and westernmost parts of the European part of Russia. The arable Chernozem (Figure 6(3)) demonstrates only plants roots on the sharp part of arable and lower horizons. These soils are spread out on the southern part of Samarskaya Luka, whitening the borders of national park, where agricultural practices are allowed with only prohibition to use mineral fertilizers. The last example of Chernozems is given in

Figure 6(4). The specificity of these soils is the color of parent materials—Akchagyl clays, blue-grey colored and vertically stratified due to transgression genesis.

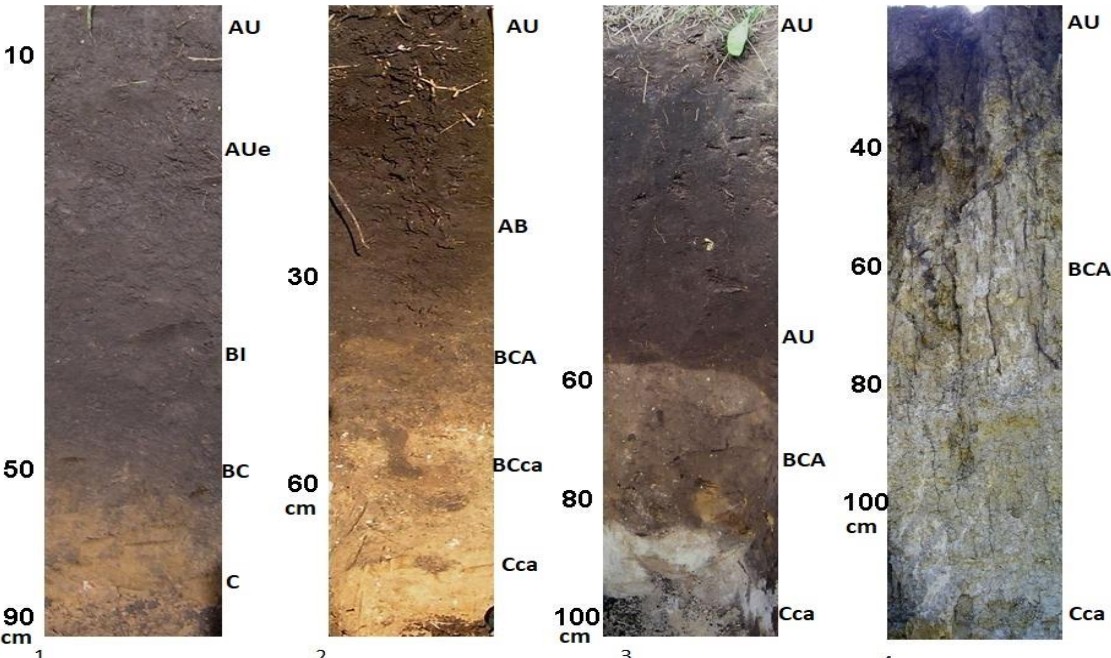

**Figure 6.** Chernozems of Samarskaya Luka. **1**—Luvic Chernozem, **2**—Calcaric Chernozem, **3**—Arable Calcaric Chernozem, **4**—Calcaric Chernozem on relic Akchagyl clays.

The floodplains of Samarskaya Luka are different in terms of hydrological regime. Thus, the northern part has less stable hydrology due to water discharge in proximity to the Kuibishevskoye water reserve catchment. The stratified Fluvisols with sharp interchange of layers are typical for this part of the peninsula (Figure 7(1)), this soil has numerous different colored layers which represent different chronological stages of soil development and river sedimentation activity. Oxbow floodplains remote from the river bed demonstrate less differentiated soils profiles (Figure 7(2,3)). The lower parts of their profiles demonstrate features of gleyification.

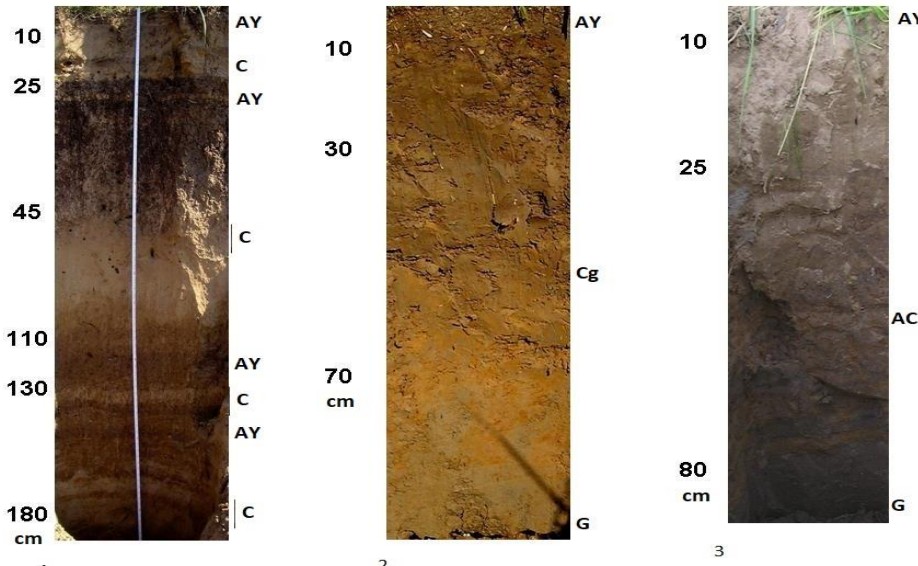

**Figure 7.** Fluvisols of floodplains on northern part of Samarskaya Luka. **1**—Fluvisol close to river course, **2,3**—Fluvisols on the oxbow floodplains.

The floodplains of the southern part of Samarskaya Luka are less affected by water discharge and pedogenesis here is more stable. Thus, alluvial processes do not appear here every year, but once every few years. Soils here (Figure 8) do not demonstrate stratified profiles with dark topsoil humus-enriched horizons of the mull type. Normally the soil of oxbow floodplains lies under forest or meadow vegetation cover. Gleyification is also pronounced in these soils' lower parts. The oxbow floodplains are very valuable as examples of old floodplains, preserved since the construction of the cascade of reservoirs. At that time, there were poplar forests, which have almost disappeared due to economic activities and coastal abrasion. Nevertheless, there are preserved areas of old floodplain soils.

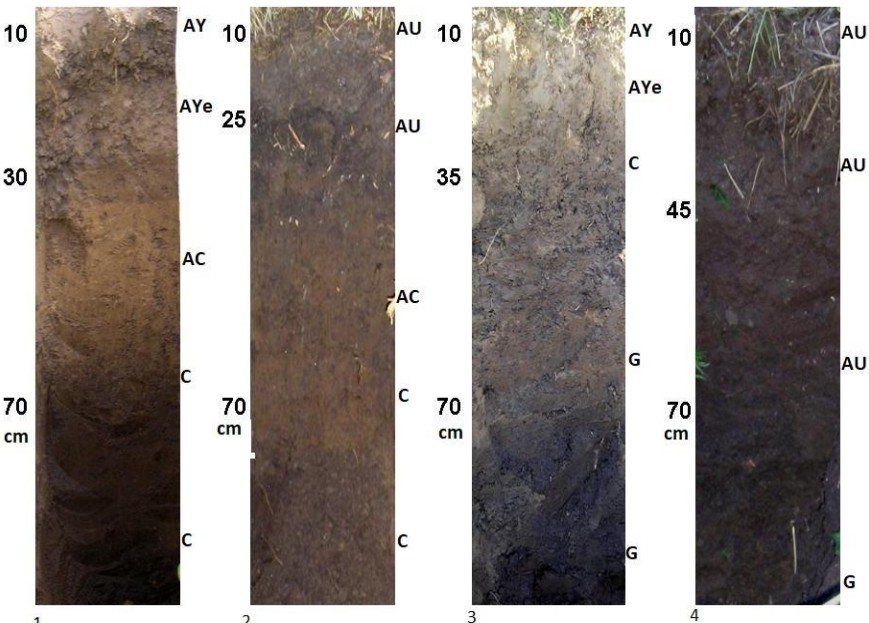

**Figure 8.** Fluvisols of floodplains on southern part of Samarskaya Luka. **1,2**—Fluvisols of old oxbow floodplains with initial features of podsolization, **3,4**—Fluvisols of old oxbow floodplains with dark humus horizons.

*3.4. General Analytical Soil Characteristics*

Basic chemical characteristic of soil as well as content of key particle size fractions are given in Table 3. Leptosols are intrazonal soils which normally form on consolidated rocks. In this study we have so-called Rendzic Leptosols formed on limestones and their derivates. That is why they have alkaline reaction in all mineral horizons, only forest floors demonstrate neutral reactions. The content of calcium increases with depth of soil as well as skeletal and sand fraction, which is typical such soil types [3]. The content of organic carbon is high as well as degree of humus enrichment by nitrogen on the basis of C/N ratio.

**Table 3.** General chemical soil characteristics and soil texture.

| Soil Type/ Horizon | TOC, % | TON, % | C/N | CaCO₃ | pH | Sand, % | Silt, % | Clay, % | Skeleton, % | Fine Earth, % |
|---|---|---|---|---|---|---|---|---|---|---|
| Leptosol | | | | | | | | | | |
| O | 39.68 | 1.20 | 32.24 | nd | 6.6 | nd | nd | nd | nd | тd |
| AU | 5.70 | 0.59 | 9.66 | 14.5 | 7.2 | 85 | 10 | 5 | 45 | 55 |
| AC | 2.47 | 0.51 | 4.84 | 22.0 | 7.8 | 89 | 10 | 1 | 57 | 43 |
| C | 0.21 | 0.03 | 7.00 | 51.2 | 8.2 | 95 | 4 | 1 | 75 | 25 |

<div align="center">

**Table 3.** *Cont*.

</div>

| Soil Type/ Horizon | TOC, % | TON, % | C/N | CaCO₃ | pH | Sand, % | Silt, % | Clay, % | Skeleton, % | Fine Earth, % |
|---|---|---|---|---|---|---|---|---|---|---|
| colspan | | | | | Cambisol Hyperskeletic | | | | | |
| O | 45.00 | 1.89 | 23.81 | nd | 6,7 | nd | nd | nd | nd | Td |
| AU | 3.20 | 0.45 | 7.11 | 3.4 | 7.2 | 35 | 35 | 30 | 68 | 32 |
| BM | 1.23 | 0.12 | 10.25 | 4.5 | 7.7 | 15 | 40 | 45 | 65 | 35 |
| BC | 0.78 | 0.05 | 15.60 | 7.6 | 8.0. | 25 | 40 | 35 | 70 | 30 |
| Cca | 0.24 | 0.03 | 8.00 | 10.2 | 8.2 | 35 | 30 | 35 | 76 | 24 |
| colspan | | | | | Cambisol | | | | | |
| O | 39.00 | 3.88 | 10.05 | nd | 7.2 | nd | nd | nd | nd | Td |
| AU | 3.90 | 0.28 | 13.92 | 0.5 | 6.8 | 20 | 40 | 40 | 10 | 90 |
| BM | 0.40 | 0.07 | 5.71 | 1.2 | 5.7 | 15 | 40 | 45 | 12 | 88 |
| BC | 0.23 | 0.05 | 4.60 | 1.7 | 6.2 | 25 | 45 | 30 | 15 | 75 |
| C | 0.15 | 0.02 | 7.59 | 4,0 | 6.8 | 30 | 35 | 35 | 18 | 82 |
| colspan | | | | | Retisol on Jurassic clays | | | | | |
| O | 46.50 | 1.55 | 30.00 | nd | 6.7 | nd | nd | nd | nd | nd |
| AY | 3.50 | 0.41 | 8.53 | 0.0 | 6.3 | 43 | 35 | 22 | 5 | 95 |
| AEL | 1.91 | 0.21 | 9.05 | 0.0 | 5.5 | 26 | 42 | 33 | 5 | 95 |
| BEL | 0.61 | 0.05 | 12.20 | 0.0 | 5.4 | 24 | 30 | 46 | 5 | 95 |
| BC | 0.50 | 0.05 | 10.00 | 0.0 | 5.2 | 35 | 16 | 49 | 5 | 97 |
| C | 0.20 | 0.03 | 6.67 | 0.0 | 5.1 | 47 | 13 | 40 | 3 | 97 |
| colspan | | | | | Retisol | | | | | |
| O | 55.70 | 2.40 | 23.20 | nd | 6.8 | 28 | 35 | 37 | nf | nd |
| AYe | 4.50 | 0.55 | 8.18 | 0.0 | 6.7 | 24 | 30 | 46 | 8 | 92 |
| BEL | 2.90 | 0.34 | 8.53 | 0.0 | 6.7 | 27 | 30 | 43 | 10 | 90 |
| BC | 1.90 | 0.22 | 8.63 | 0.5 | 6.9 | 27 | 31 | 42 | 10 | 90 |
| C | 0.50 | 0.06 | 8.33 | 1.2 | 7.2 | 17 | 28 | 55 | 15 | 85 |
| colspan | | | | | Chernozem Luvic | | | | | |
| AU | 5.60 | 0.45 | 12.44 | 0.0 | 6.0 | 22 | 32 | 46 | 12 | 88 |
| AUe | 3.14 | 0.38 | 8.26 | 0.0 | 6.0 | 21 | 30 | 49 | 10 | 90 |
| BI | 2.32 | 0.35 | 6.62 | 0.0 | 5.9 | 17 | 35 | 48 | 8 | 92 |
| BC | 1.70 | 0.25 | 6.80 | 0.5 | 6.0 | 32 | 18 | 50 | 8 | 92 |
| C | 0.50 | 0.07 | 7.14 | 1.5 | 6.3 | 33 | 16 | 51 | 10 | 90 |
| colspan | | | | | Chernozem Calcaric | | | | | |
| AU | 7.89 | 0.98 | 8.05 | 1.5 | 7.2 | 23 | 29 | 48 | 15 | 85 |
| AB | 6.78 | 0.66 | 10.27 | 2.6 | 7.4 | 23 | 26 | 51 | 15 | 85 |
| BCA | 3.45 | 0.45 | 7.66 | 2.9 | 8.0 | 25 | 22 | 53 | 18 | 82 |
| BCca | 2.13 | 0.23 | 9.26 | 7.5 | 8.5 | 21 | 29 | 50 | 19 | 81 |
| Cca | 0.54 | 0.07 | 7.71 | 8.0 | 8.5 | 22 | 29 | 49 | 20 | 80 |
| colspan | | | | | Fluvisols close to river bed | | | | | |
| AY | 2.34 | 0.23 | 10.18 | 0.0 | 5.9 | 31 | 44 | 25 | 10 | 90 |
| C | 1.21 | 0.17 | 7.11 | 0.0 | 6.0 | 45 | 37 | 18 | 19 | 81 |
| AY | 2.45 | 0.25 | 9.80 | 0.0 | 6.2 | 29 | 45 | 26 | 8 | 92 |
| C | 0.56 | 0.09 | 6.22 | 0.0 | 6.0 | 49 | 32 | 19 | 26 | 74 |
| AY | 3.45 | 0.31 | 11.13 | 0.0 | 5.8 | 23 | 47 | 30 | 11 | 89 |
| C | 0.23 | 0.03 | 7.66 | 0.0 | 5.9 | 52 | 31 | 17 | 32 | 68 |

**Table 3.** *Cont.*

| Soil Type/ Horizon | TOC, % | TON, % | C/N | CaCO$_3$ | pH | Sand, % | Silt, % | Clay, % | Skeleton, % | Fine Earth, % |
|---|---|---|---|---|---|---|---|---|---|---|
| | | | | Fluvisol of oxbow flood plains | | | | | | |
| AU | 5.67 | 0.34 | 16.67 | 0.0 | 6.8 | 20 | 45 | 35 | 5 | 95 |
| Cg | 1.00 | 0.08 | 12.50 | 0.0 | 6.9 | 19 | 43 | 38 | 3 | 97 |
| G | 0.43 | 0.05 | 8.60 | 0.0 | 7.0 | 18 | 44 | 38 | 10 | 90 |

Cambisols or brown forest soils are a very broad group of soils, this term and its content have long evolved. Currently, the key feature of Cambisols is sharp change (Spanish word "cambio") of texture and mineralogy in metamorphic horizon BM. Both soils of this type demonstrate changes in texture in BM in Russian soil taxonomy [20] or Bw in WRB layers. Superficial horizons are enriched by organic carbon, while its content in mineral middle-layer becomes essentially lower, which is also quite typical for this soil type [19]. The content of fine earth higher in the Cambisol is located on the middle part of slope if one compares it with Hyperskeletic Cambisol on the border with polypedones of Leptosols. The skeletal fraction is a source of calcium for fine earth, that is why skeletal soil is normally more carbonate.

Retisols are very broad group of soils [19] with texture differentiation of soil profile due to development of eluvial-illuvial processes. Our soils located in subboreal climate with seasonal lack of precipitation, that is albeluvic properties, are not as abundant as in Poldzolic soils of the taiga. Nevertheless, thickness of eluvial and illuvial layers higher in the soil formed in Jurassic clays, which are acidic and free of carbonates. In the case of low carbonate deluviums the intensity of texture differentiation is lower, soil is more dark-colored and pH values are shifted to the alkaline ranges. In general, our soils are comparable with Retisols of the nearby Privolzhskaya Upland [4].

Chernozems from Samarskaya Luka are presented by two groups—Luvic Chernozems and Calcaric Chernozems. The first group is associated with leached loams and clays. The second one normally forms on carbonate containing loess-like parent materials. Thus, the alkalinity of the parent material plays a crucial role in spatial differentiation of chernozemic soil cover. Luvic Chernozems are less alkaline, have less organic matter content and demonstrate some textural differentiation of the middle part of the soil profile. Our Chernozems are essentially enriched by humus in topsoil and middle parts of the solum. At the same time, the thickness of the organic layer is less than in the same soils in the Srednerusskya upland [9,21,22].

Fluvisols are different in terms of chemical and particle size distribution. Thus, young polygenetic Fluvisols located in river beds are very inhomogeneous in vertical scale, which indicate changes of the soil formation stages within the period. This is typical in the lithogenic sequence of soil layers, where humus horizons, buried by alluvium, represent polygenetic types of pedogenesis. Vertical inhomogeneity is expressed in the texture, organic carbon and nitrogen in this soil section. Those soils that have long since emerged from the hollow regime do not demonstrate layer by layer stratification as in alluvial soil. In both alluvial soils the content of organic matter is high and this designates their role in accumulation and stabilization of organic matter.

*3.5. Soil Spatial Distribution*

Soil catena is presented in Figure 9. This catena is oriented from north (Volga River) to south. It is evident that soil cover is quite inhomogeneous and relief plays a crucial role in spatial soil differentiation. Such detailed soil catena was created for the forest-steppe part of Samarskaya Luka, because it is located in the Zhiguly state reserve territory and are not affected by anthropogenic influences. The soil catena starts from mountain soils— Cambisols and Leptosols. The Leptosols from the top of mountain ridges are hyperskeletic.

Many of the mountain soils are classified as inclinic, because they are located on the steep slopes. Zhiguli mountains were changed by lowlands of the late Pleistocene age, formed under long denudation and erosion. The relief heights here are lower, in some cases lower than 100 m, and soils of Retisol type and Chernozem Luvic are typical for this environment. The appearance of carbonate parent materials on the southern uplands result in formation of Calcaric Chernozems, while the presence of dense clays affect soil over moistening and pronounced gleyic or gleyified soils.

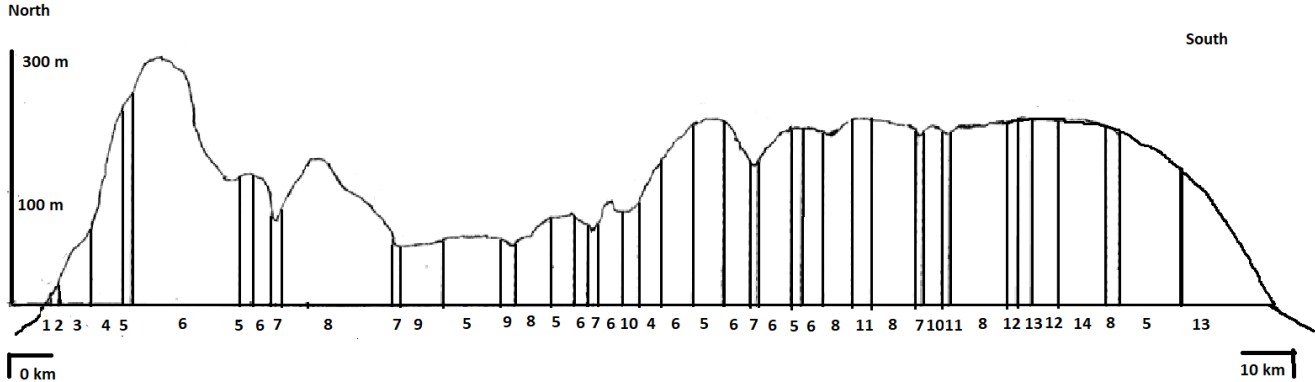

**Figure 9.** Soil catena from north to south of Samarskaya Luka. **1**—river bank rocky shoal, **2**—Leptosol Calcaric, **3**—Retisols, **4**—Cambisol Luvic, **5**—Leptosol Calcaric Cambic, **6**—Leptosol Calcaric Hyperskeletic, **7**—Stratified sediments of mountain foothills, **8**—Retisol Umbric, **9**—Chernozem Luvic, **10**—Retisol, **11**—Chernozem Calcaric, **12**—Gleysol, **13**—Retisol Gleyic, **14**—Retisol Argic.

The schematic visualization of soil distribution of Samarskaya Luka is given in Figure 10.

This figure illustrates differentiation of soil cover, namely, the northern part of the territory presented by Cambisols and Leptosols, including Leptosols on gypsum rocks. The last soils are very localized in former gypsum mines in the eastern part of Samarskaya Luka, which were abandoned more than 200 years ago. Thus, this soil could be considered as natural. Similar soils with low developed profiles were described by Goryachkin et al. [23] for the northern boreal regions of European Russia. Thechnosols are found in the northern mountain area, where limestone mining was recently active or is currently ongoing. Antrosols are typical for settlements—cities and villages—and they are neighbors to agrogenic versions of local soils (Retisol, Chernozems, etc.). Two of the largest soil areas are presented by Retisols and Chernozems Calcaric. They extend from west to east and indicate the border between forest-steppe and normal steppe. This corresponds well with zonation of the region, published by E. Kolomyts [8]. Data obtained in this research will fill the gap in spatial soil science and modeling [24], which are characterized by initial data scarcity. Our data also support an idea that lithology regulates key soil properties via alteration of the soil texture [25]. Our study has shown that carbonate content in parent materials results in dividing of soil cover into Retisols and Chernozem, which exist in adjacent polypedones. At the same time, calcium content could be a key predictor of spatial transfer of Leptosols to Chernozems, which was reported recently for the Carpathian mountains [26]. The origin of Cambisols and the role of local climate in the mountain regions of Povolzhye has also been clarified in this study and this corresponds well with data of Kostenko, derived for the Crimean Mountains [27]. Furthermore, our study found that soil-forming parent materials play a crucial role in soil formation not only in cold and humid regions [28], but also in semi-arid environments, where they can determine the morphogenesis of the zonal soil type and the very differentiation of the soil cover into zones. This manuscript summarized the data of field studies conducted earlier and allowed us to identify the relationship of the main soil types to the parent materials. The role of geogenic factors in the formation of zonal gradient of soils in the transboundary territory was revealed for the Central Volga region.

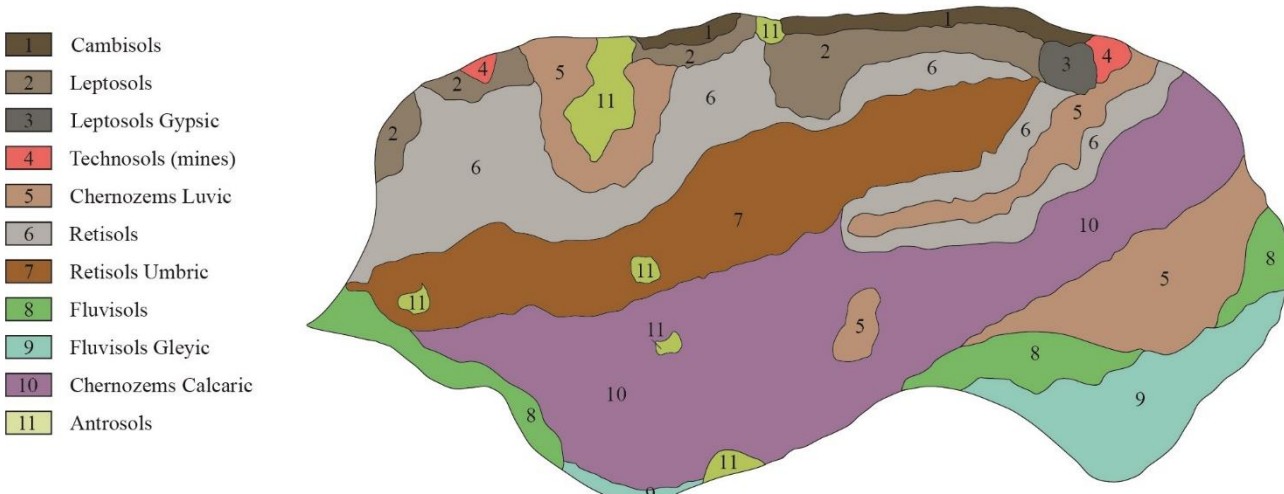

**Figure 10.** Schematic soil map of Samarskaya Luka.

## 4. Conclusions

The soil cover of Samarskaya Luka is very heterogeneous and diverse. The diversity of soil types is primarily determined by geogenic conditions. Thus, the soil cover of Samarskaya Luka is characterized by considerable contrast and heterogeneity along its entire length. This type of soil cover structure belongs to the category of lithologically differentiated, in which differences between components are determined by initial heterogeneity of soil-forming rocks (heterogeneity of lithological composition, different depth of occurrence of dense carbonate rocks). Adjacent landscape areas are characterized by less contrasting soil cover. Thus, in the northern part, sandy deposits with pine forests dominate. In the west, there is a flatter terrain with zonal Retisols and Luvic Chernozems. On the east, red-colored parent materials appear on uplands and specific red-colored soils dominate. As for the southern direction there are lowlands with salinized and gleyified soils. Thus, the lithological factor at the center of a series of latitudinal zonation radically changes the diversity of soils within the forest-steppe zone. The soil-forming potential of the environment significantly changes through geogenic factors, and also influences the biotopic diversity.

**Funding:** This research received no external funding.

**Institutional Review Board Statement:** Not applicable.

**Informed Consent Statement:** Not applicable.

**Data Availability Statement:** (1) Data available in a publicly accessible repository. The data presented in this study are openly available in [repository name e.g., FigShare] at [doi], reference number [reference number].

**Acknowledgments:** Great acknowledgements are made to Elvira Gagarina who initiated this research in 2001, to Vladimir Vechnik, Zhigulevsky State Reserve for help with field work and for many students of Saint-Petersburg State University for their work in expeditions. The article is in honor of the 300th anniversary of St. Petersburg State University. This manuscript is dedicated to the memory of Sergey Saxonov, Zhigulevsky State Reserve and Institute of Ecology of Volga Basin, who passed away during the pandemic of COVID-19.

**Conflicts of Interest:** The authors declare no conflict of interest.

## Abbreviations

| | |
|---|---|
| WRB | world reference base of soil resources |
| O | forest floor |
| AU | dark humus horizon |
| AY | light humus horizon |
| AUe | dark humus horizon with eluviation of clay |
| AYe | light humus horizon with eluviation of clay |
| AEL | humus eluvial horizon |
| AB | transitional layer between A and B |
| AC | transitional layer between A and C |
| BEL | sub-eluvial horizon |
| Bw | weathering soil layer |
| Bm | cambic soil horizon |
| BI | clay illuvial horizon |
| BCA | horizon of secondary accumulation of carbonates |
| BCca | transitional horizon with primary carbonates |
| TOC | total organic carbon |
| TON | total organic nitrogen |
| C/N | carbon to nitrogen ration |
| C | parent material |
| G | gley horizon |
| Cg | gleyic horizon |

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
