# Peer review of "Pedodiversity of Subboreal Ecosystems under Contrasting Geogenic Factors (Case Study of Samarskaya Luka, Middle Volga Region, Russia)"

_geosciences, doi:10.3390/geosciences12120443_

Round 1
Reviewer 1 Report
I carefully analyzed the paper. The topic is of great interest, the zonation of soils, as a result of the solicification and environmental factors, being an interesting and sensitive topic.
Although soil science in Russia has a special history, the paper does not fit into this situation, being without adequate documentation, and especially without accuracy, soil pictures are inadmissible.
In order for the paper to be publishable and to meet the minimum conditions of a scientific paper, I believe that the following improvements are necessary:
1. Improving the scientific documentation of the paper with relevant references at the national and international level. Then specifying the research hypothesis and emphasizing the novelty of the research carried out within the paper.
2. Insert a sub-chapter on Material and research method - where you specify which research methods you used with appropriate references.
3. Use pictures with soil profiles in which the highlight of the horizons is integrated in the picture, to be real from the field (don't put the depths on the edge randomly).
4. Specify what all the abbreviations in the tables mean in the footnotes.
Author Response
Dear reviwer! thank you for very usefull comments and suggestions. The text of manuscript has been rewoked. More international papers has been cited in the discussion chapter. Special chapter on soil spatial distribution with schematic map has been added. Materials and methods chapter has been expanded. Pictures of soils has been reworked. Sincerely yours, Evgeny Abakumov
Reviewer 2 Report
Dear Authors,
I have read your manuscript with great interested and I was looking forward to see some real soil diversity supported by numerous great maps. After reading the manuscript I didnt get what I was expecting. The manuscript is interesting and certainly has great value but to have this work published in an international journal, substantial revision and editing is inevitable. There are sections where I find really difficult to understand what I supposed to follow. There are also large number of long and difficult to follow sentences not really helping for the reader. The manuscript totally underutilized the power of good figures and most importantly, maps. As the manuscript subject is soil diversity, some sort of spatial representation of the data is a must. There are large number of locations mentioned, commonly referred to some soil diversity elements, hence without proper maps the reader just got lost. This need to be fixed. Many of the soil figures are promising, but currently no real figure caption associated with them hence the reader not sure what to look for or how to compare them. There are also several technical problems and style issues that cannot appear in a top level manuscript. I have provided an annotated PDF file where you can see the exact places I find some issues to resolve. Please consider those comments and complete the revision.
Kind regards,

Author Response
Dear reviwer, thank you for usefullcomments. The manuscript has been essentially reworke accoridng you comments to pdf file. Namely, some paragraphs were rewritten, more international references were added, figure are reworked and additional figures regarding soil spatial distributio were added. Additional chapter about soil spatial distribution was added to the manuscript text. Sincerely yours, Evgeny Abakumov
Round 2
Reviewer 1 Report
The paper has been significantly improved.
I recommend accepting the paper for publication.
Author Response
Dear reviwer, thank you for your comments, english spelling and minor changes has been provided in text.
Reviewer 2 Report
Dear Authors,
I have checked your revised manuscript and find that there is no doubt you considered most of the comments and fixed them in good level. I think the manuscript is in good shape now and nearly acceptable, only minor revision is needed. Below I list few issues you could fix easily.
1) While the manuscript language is more or less okay by now, still, if you can, a reading by a native speaker would really improve the quality of the manuscript.
2) In Figure 3, please expand the figure caption as you have three frames and the reader need to be guided what they see on them. It is not enough that you just have the story in the main text.
3) The above point is valid for Figures 4, 5, 6, 7 and 8.
4) Check the tables, like Table 2 as the label words messed up in few cases.
5) On Figure 10, could you please add a zebra frame with coordinates to the map as well as a North arrow.
I think the manuscript otherwise is nice and nearly ready to be published.
Best regards,
Author Response
Dear editor!
thank you for you comments. all of your comments have been taken into account, and the text of the article has been amended accordingly.
figures and tables were improved.